# Challenges and Opportunities for the Translation of Single-Cell RNA Sequencing Technologies to Dermatology

**DOI:** 10.3390/life12010067

**Published:** 2022-01-04

**Authors:** Alex M. Ascensión, Marcos J. Araúzo-Bravo, Ander Izeta

**Affiliations:** 1Tissue Engineering Group, Biodonostia Health Research Institute, 20014 Donostia-San Sebastián, Spain; alex.ascension@biodonostia.org; 2Computational Biology and Systems Biomedicine Group, Biodonostia Health Research Institute, 20014 Donostia-San Sebastián, Spain; marcos.arauzo@biodonostia.org; 3Max Planck Institute for Molecular Biomedicine, 48167 Muenster, Germany; 4IKERBASQUE, Basque Foundation for Science, 48012 Bilbao, Spain; 5School of Engineering, Tecnun-University of Navarra, 20009 Donostia-San Sebastián, Spain

**Keywords:** single-cell, skin, cell population heterogeneity, dermatological disease

## Abstract

Skin is a complex and heterogeneous organ at the cellular level. This complexity is beginning to be understood through the application of single-cell genomics and computational tools. A large number of datasets that shed light on how the different human skin cell types interact in homeostasis—and what ceases to work in diverse dermatological diseases—have been generated and are publicly available. However, translation of these novel aspects to the clinic is lacking. This review aims to summarize the state-of-the-art of skin biology using single-cell technologies, with a special focus on skin pathologies and the translation of mechanistic findings to the clinic. The main implications of this review are to summarize the benefits and limitations of single-cell analysis and thus help translate the emerging insights from these novel techniques to the bedside.

## 1. Introduction

Skin represents the outermost barrier against foreign objects, radiation, and other insults [1]. It also acts as a thermal insulator and regulator and senses external stimuli. To accommodate these varied functions, the cellular composition of skin is very heterogeneous, including numerous cell types [2]. In further rounds of complexity, epidermal adnexa are composed of additional specialized cell types, and each skin cell type will be represented in a given time by a variable number of cell states that respond to minor alterations of the cell microenvironment.

Traditionally, three keratinocyte-basal, suprabasal, and corneocyte-and two fibroblast-papillary and reticular- subtypes have been distinguished based on differential localization within layers, as well as discrete changes in gene and protein expression [3,4]. Recently, single-cell analysis methods have captured this heterogeneity more systematically. Single-cell studies focus on the biological properties of each cell after tissue disaggregation, i.e., the status of genome, transcriptome, proteome, or epigenome may be described within each dissociated cell. In contrast, the traditional bulk analysis permits analyzing properties of groups of cells or tissues as a whole, thereby losing cell-intrinsic granularity. Single-cell studies have shown that gene expression is variable even in similar cell types [5], broadening the paradigm of cell heterogeneity. Instead of thinking of cell types, biologists now more commonly refer to cell states, i.e., slight changes of cell identity in response to the environment that result in several subpopulations included within a broader cell type definition. More than 1000 single-cell datasets are publicly available [6] and the number is increasing by the day. Therefore, it is conceivable that cell types and states will be redefined to varying degrees of complexity depending on the resolution used in computational clustering [7].

Depending on the number of measured biological features, single-cell methods are divided into unimodal and multimodal if one or more than one feature—e.g., transcriptomic expression or epigenomic signature—are evaluated [8,9,10]. The prevalent unimodal method is single-cell RNA- sequencing of dissociated cells [11], which captures the gene expression profile of each cell. Methods that measure the epigenomic profile of cells are less common, mainly due to the complexity of the protocols and the relatively low level of information obtained. Multimodal methods vastly increase the cost and complexity of experiments, thus impeding wide adoption by the research community [10].

A prominent field within the multimodal methods are fields of single-cell protein detection methods. These methods can be divided into two families. Mass spectrometry(MS)-based single-cell methods couple single cell isolation with MS [12,13]. While those methods show a high resolution proteomic profile (>1000 peptides) their cell throughput is scarce (∼10–100 cells) and lack robust data integration pipelines [14]. The second family of methods are based on cell surface protein analysis, in which antibodies against membrane proteins are coupled with DNA tags, which are sequenced as the rest of elements within the droplet [15,16,17]. While those methods have a higher cell throughput (∼10 k) and are interpreted as a wider version of FACS, the array of antibodies to be used—about 100, biased towards immune populations—is far from a golden standard for general tissue analysis.

More recently, spatial transcriptomic methods have been developed [18,19,20]. These methods combine histology sampling with RNA-seq or proteomic signatures of individual cells or small regions within the tissue sample. With these methods, the expression signatures of hundreds to thousands of genes are linked to specific regions within the tissue, thus providing relevant spatial information. Currently, two main types of spatial methods exist. In in situ hybridization and sequencing methods such as *seqFISH+* [21], *MERFISH* [22] or *STARmap* [23], cell mRNAs are hybridized with fluorescent probes designed for an array of genes, and the location of the probe is identified using a sequencer. On the other hand, in situ mRNA capturing methods such as Slide-seq [24] or *Visium* [25] use custom slides with immobilized probes, set up to capture spatial information. Then, tissue sections are deposited on the slides, and the mRNA molecules of the tissue are hybridized to the probes on the slide, amplified, and sequenced. The former set of methods allows subcellular resolution of gene expression but do not capture the whole transcriptional landscape of the cell—although more recent techniques such as *seqFISH+* allow for quasi-transcriptional levels—whereas the latter shows a more representative picture of the transcriptomic status of the selected area, but section areas tend to include several cells, and thus, transcriptomic data do not present single-cell resolution. However, novel iterations of available commercial platforms are claiming near subcellular resolution, indicating that the technological challenges for improving spatial resolution are being solved.

In this review, we recapitulate the main cell populations and subpopulations observed by single-cell RNA-sequencing (scRNA-seq) technologies in healthy and pathological skin focused on human skin with limited reference to animal models. We enumerate current challenges in single-cell methods and the lessons learned for experimental design. Lastly, we summarize the opportunities that single-cell technologies will bring to interested dermatologists and skin biologists. Our approach complements previous reviews focused on scRNA-seq techniques in dermatology [26,27,28,29,30] in that we explore the actual cell populations and pathways in depth. We also aim to increase awareness of the exploding potential that these techniques make available to clinical medicine and, ultimately, highlight their potential impact in the development of novel diagnostic and treatment options for dermatological diseases.

## 2. Single-Cell Analysis of Healthy Skin

Any cellular analyses in humans depend on relatively scarce live tissue availability. Additionally, experimental protocols hugely impact data because early injury response genes light up in response to tissue disaggregation [31]. Of all solid tissues, the study of skin has gained traction in the single-cell community because it is accessible and readily available. However, skin disaggregation—as with any other solid tissue—presents some technical challenges, and this will impact cell viability, as well as the representativeness of the obtained sample [32,33]. To date, most of the single-cell studies on healthy skin are focused either on determining the heterogeneity of fibroblasts from the dermis or understanding transcriptional changes underlying the differentiation of keratinocytes in the interfollicular epidermis. However, the skin is a highly specialized organ with several regional specifications. Of interest, a recent study analyzed specialized cell populations of the nails [34]. Hopefully, similar studies will follow suit to better understand all skin appendages.

### 2.1. Single-Cell Analyses of Fibroblasts

Fibroblasts are the main cellular constituent of the dermis. Histologically, the dermis presents distinct upper (papillary) and lower (reticular) layers (Figure 1). Papillary fibroblasts are more densely packed, while the reticular dermis contains sparse fibroblasts and substantially more extracellular matrix (ECM). For a long time, phenotypic markers that distinguish both fibroblasts types have been explored. However, it is now clear that both fibroblast categories contain subpopulations [35], and more granularity is thus needed to achieve a biologically meaningful categorization of cell subsets.

Several research groups have now proposed diverse clusterings of human skin fibroblast populations at the single-cell level, with all published studies finding three major cell populations and a varying number of subpopulations [36,37,38,39,40]. Unfortunately, there seems to be little overlap between studies concerning markers of fibroblast types and subtypes. This is only apparent because in a joint reanalysis of the first four datasets high comparability between them was uncovered [35]. Thus, there seem to be 3 major fibroblast types in human skin, denominated A-C, and they are composed of at least 10 minor subtypes (clusters), denominated A1–A4, B1 and B2, and C1–C4. For instance, Ascension’s cluster B2 (*CCL19*${}^{+}$) can be mapped to Tabib’s cluster 5 [36], Solé-Boldo’s 2A population [38], and Vorstandlechner’s cluster 2 [37]. Similarly, Ascension’s cluster A2 (*APCDD1*${}^{+}$*COL18A1*${}^{+}$) can be mapped to Tabib’s cluster 0, Solé-Boldo’s cluster 3, and Vorstandlechner’s cluster 3. These computationally determined axes and clusters account for 92.5% of the sequenced fibroblasts. There are some indications of the localization of some of these cell types (Figure 1), but the functional characterization of fibroblast types and subtypes is still lacking. Recently, a large (>0.5 M cells) sc-RNAseq dataset (again including three major fibroblast subpopulations on human skin) was made available. In this case, fibroblast subpopulations were characterized by the expression of signature genes *COL1A1* and *COL1A2*, *CXCL12*, and *CCL19*, respectively [40]. Of note, a preprint posting a reanalysis of this study postulates that their fibroblast clustering is unreliable due to technical reasons [33].

All previously mentioned studies focused on human skin. However, fibroblasts are ubiquitous cells, and data from animal models as well as data on fibroblasts from other organs may yield interesting insights. For instance, Buechler et al. performed a single-cell RNA-seq analysis of 120,000 fibroblasts obtained from different mouse tissues, including the skin [41]. In the healthy mouse tissues analyzed, they found two major pan-tissue “universal” fibroblast types that reliably appeared across most tissues, namely, a vascular niche-associated *Dpt*${}^{+}$*Pi16*${}^{+}$ population, with stem cell-like characteristics, and a basement membrane-associated *Dpt*${}^{+}$*Col15a1*${}^{+}$ population. They also described other minor “specialized” fibroblast populations such as *Ccl19*${}^{+}$, *Coch*${}^{+}$, *Comp*${}^{+}$, *Cxcl12*${}^{+}$, *Fbln1*${}^{+}$, *Bmp4*${}^{+}$, *Npnt*${}^{+}$, and *Hhip*${}^{+}$ cells. To the naked eye, some of these populations may seem transcriptomically related to the above-described human skin fibroblast clusters [35]. However, we find little replication of mouse fibroblast markers in the human populations (Appendix A), indicating that care should be taken in assuming any correlation between mouse and human studies. Some facts may lead to potentially interesting discoveries. For instance, *Coch*${}^{+}$ fibroblasts have also been described in other human organs [42]. Thus, in-depth comparative studies of the human skin fibroblast populations and other organ and species datasets may prove a fruitful line of research to start disentangling functional cell types within human dermal fibroblasts.

### 2.2. Single-Cell Analyses of Keratinocytes

The interfollicular epidermis is composed of four keratinocyte layers, showing a continuum of differentiation: the basal stratum, located just above the basement membrane between the dermis and epidermis, is the least differentiated state. Cells from this layer differentiate into the spinous, granular, and cornified layers, where they gradually lose their nuclei and increase their keratin content. Due to their low amount of expressed RNA, cells from the granular and cornified layers are rarely captured during single-cell analysis. As expected, the greater complexity of these layers has only been appreciated by the increased granularity of scRNA-seq studies.

The first study by Cheng et al. researched the heterogeneity of epidermal cell types on the human scalp, trunk, and foreskin tissue [43]. All skin sources shared at least three keratinocyte populations: spinous, granular, and follicular. They also identified a mitotic population with high *PCNA* and *KI67* expression, and a channel population expressing cell junctions (*GJB2*, *GJB6*) and mitochondrial channels (*VDAC2*). They found two populations specific to the scalp: a follicular population (*S100A2*, *APOE*) and a WNT inhibitor population, which likely constituted hair follicle bulge cells (*SFRP1*, *FRZB*, and *DKK3* positive). Finally, they detected two basal keratinocyte subpopulations, one present in trunk and scalp skin (*CXCL14*${}^{\mathrm{hi}}$, *DMKN*${}^{\mathrm{hi}}$) and the second one in foreskin (*CCL2*${}^{\mathrm{hi}}$, *IL1R2*${}^{\mathrm{hi}}$) which exclusively expressed amphiregulin. A later reanalysis of the neonatal foreskin data by the same group recapitulated the epidermal differentiation into 7 discrete stages [44], characterized by different Gene Modules. For instance, module 4 consisted of calcium-binding and cell adhesion genes (*CDH3*, *FAT1*, *DSG3*); module 2 was enriched in mitotic stage-associated keratins (*KRT6A*, *KRT6B*); and module 5 contained genes involved in barrier function (*DEGS2*, *CERS3*), cell adhesion (*DSC1*, *PERP*), tight junctions (*CLDN1*, *CLDN8*), and desquamation (*KLK8*, *KLK11*). A mitotic population showed a high expression of basal markers and intermediate expression of early differentiation markers (*KRT10*, *KRT1*). Finally, they identified *ETV4* and *ZBED2* as key transcription factors to maintain the basal keratinocyte state.

A second study by Wang et al. detected 4 basal keratinocyte populations—two of which were transitional—in neonatal epidermis, 3 spinous populations, and one granular population, in addition to melanocytes and Langerhans cells [45]. When reconstructing the differentiation trajectory of keratinocytes, they developed an estimate of the likelihood of one cell type transitioning to another. For instance, they observed that basal III and IV communities had the highest likelihood to transition to spinous I. They also observed that the expression of *PTTG1* in basal keratinocyte I and *HELLS* and *UHRF1* in basal keratinocyte II were required for epidermal homeostasis. Concerning putative stem cell populations, it seems that spatial location may be more relevant than previously thought, and the authors show the importance of the localization of the different cell populations in relation to epidermal rete ridges. Of note, scRNA-seq analyses of clonogenic keratinocytes in culture showed that the holoclone-forming cells are defined by the expression of *FOXM1* gene downstream of *YAP* [46].

With regard to epidermal appendages such as the hair follicle, the information from mouse models may contribute some light on the putative functions of the different cell subpopulations described. In a pioneering study, Joost et al. [47] found 25 cell populations, some of them previously not characterized like an upper hair follicle population expressing *Rbp1*, *Defb6*, and *Cst6*, located at the sebaceous gland opening. They computationally recreated the differentiation of keratinocytes from the interfollicular epidermis (IFE), and the spatial location in the proximal-distal axis of most of the populations in the IFE and HF. Finally, they observed that based on classical markers of stem cells progenitors (SCPs) like *Cd34*, *Lgr5*, *Lgr6*, or *Lrig1* those markers were not sufficient to delineate basal cell populations, because up to 33% of suprabasal cells expressed those markers, and up to 27% of SCM did not express any of these markers. A similar study performed by Takahashi et al. in human samples [48] replicated the IFE differentiation pattern, from *KRT5*${}^{+}$*KRT14*${}^{+}$ basal cells to fully-differentiated *KRT10*${}^{+}$*CALML5*${}^{+}$ cells. They also found a mitotically active subset of cells, previously detected by Cheng et al. [43]. In mouse IFE, *GRHL3* seems to control stemness at the basal cell compartment [49].

## 3. Single-Cell Technology Applied to Skin Conditions

Most dermatological single-cell studies have not focused on healthy human skin. They have rather explored the applicability of these techniques to explore pathological conditions. Because single-cell allows the detection of cell states and variations across samples, we can translate these techniques into the detection of subpopulations that appear, disappear, or change during a disease process. Focusing on skin pathologies, cancer has perhaps been most extensively studied using single-cell analysis [50,51]. As usual, data from animal models may also shed some light on human pathology. For instance, the aforementioned study by Buechler et al. described cell subsets that were specific to “perturbed-state tissues”, including the skin, such as *Cxcl5*${}^{+}$, *Adamdec1*${}^{+}$, and *Lrrc15*${}^{+}$ fibroblasts [41]. Similarly, a recent preprint study proposes human *CXCL10*${}^{+}$*CCL19*${}^{+}$ immune-interacting fibroblasts and *SPARC*${}^{+}$*COL3A1*${}^{+}$ vascular-associated fibroblasts as shared “pathogenic activation states” in tissue fibroblasts across four chronic inflammatory diseases affecting diverse tissues [52].

We will now summarize some of the most relevant single-cell studies focused on a range of dermatological diseases and skin-related phenotypes.

### 3.1. Aging of Human Skin

Although aging is a generalized process affecting all organs and tissues, the study of aged skin and the search for potential rejuvenating mechanisms are of specific relevance for the cosmetic industry. Single-cell studies of both mouse [53] and human [38,54] skin have reported that old skin samples seem to present a less-defined transcriptomic signature, indicating a loss of cellular identity. Thus, fibroblast subpopulations acquire a low-inflammatory chronic status, and their functional phenotype is less well-defined, e.g., ECM component secretion is more variable across subpopulations, compared to samples from young individuals. Additionally, Solé-Boldo et al. observed a reduced peroxide metabolism, under a lower metabolic profile [38]; and Salzer et al. detected an increase in adipogenic signaling activation, which is supported by a thicker hypodermis layer in older mice [53]. However, the latter finding is most likely specific to mouse skin [55]. Finally, Zou et al. found increased inflammation and decreased self-renewal as hallmarks of aged cells. More specifically, they found matrix disassembly genes (*MMP2*) in fibroblasts and downregulation of DNA repair genes [54]. Ligand-receptor interaction analysis revealed that *JAG1/DLL1-NOTCH3* interaction FB1-EC decreased over age, suggesting the involvement of *NOTCH-HES1* axis in the maintenance of skin homeostasis. Knockdown of *HES1*, *IER2*, *ID3*, or *TSC22D1* promoted senescence of fibroblast cell lines.

### 3.2. Atopic Dermatitis (AD)

Atopic dermatitis studies based on bulk microarray, GWAS and NGS methods have determined key points in the molecular aspects of the disease [56,57,58]. First, differentiated keratinocytes switch from a granular and corneum profile to a more basal state by *FLG* and *LOR* downregulation, as well as downregulation of genes associated with the lipid barrier. Second, Th17-associated and innate immune responses are present (*IL23*, *CCL19*). The most relevant cytokine is IL13, as compared to the IL17A-driven response in psoriasis [59].

The more recent single-cell studies have switched the focus from the epidermal keratinocytes to a more detailed analysis of immune populations. Thus, He et al. [39] found immune cell subpopulations uniquely described in AD samples, such as *LAMP3*${}^{+}$*CCR7*${}^{+}$ and *CD1A*${}^{+}$*FCER1A*${}^{+}$ dendritic cell subpopulations, a *CD68*${}^{+}$ macrophage, and *CD69*${}^{+}$*CD103*${}^{+}$*CD8*${}^{+}$ T cell populations. Rojahn et al. [60] found dendritic cells and macrophages overexpressing inflammatory markers *RNASE*, *LYZ*, *CCL17* and *AREG*—which promotes keratinocyte proliferation—as well as ECM degradation markers *CDH3*, *PLAU*, *CD40*, and *TNFRSF9*. They also discovered an increase of several T cell populations expressing *IL13*, *IL22*, *GZMB*, and *NKG7*. The latter findings were replicated by Reynolds et al. [40]. In summary, AD is a complex condition where fibroblasts, immune cells and keratinocyte responses are tightly intertwined (Figure 2).

### 3.3. Cutaneous T-Cell Lymphoma (CTCL)

CTCL produces chronic inflammation and accumulation of malignant T lymphocytes in the skin. Patients present erythroderma, lymphadenopathy and circulating T cells, as well as mycosis fungoides, in which malignant cells reside primarily in the skin. Gaydosik et al. [61] used scRNA-seq to profile advanced-stage CTCL skin tumor samples. They discovered a minimal T cell overlap between CTCL and control samples, which expressed genes associated with mTOR signaling, NK receptors, tumor cell survival, S100 and galectin families. This transcriptomic signature indicates skin barrier inflammation and dysfunction, and increased cell proliferation, motility and invasiveness.

### 3.4. Drug Reaction with Eosinophilia and Systemic Symptoms (DRESS)

DRESS is a systemic skin hypersensibility syndrome. Kim et al. [62] performed scRNA-seq on skin and blood from a patient with DRESS and identified JAK-STAT signaling pathway as a potential target, consistent with the dense infiltration of CD4${}^{+}$, and CD8${}^{+}$ T cells. They found an enrichment of pathways regarding lymphocyte activation and signaling such as *IL2RG*, *JAK3*, and *STAT1*, as well as genes involved in cell proliferation and migration (*MKI67*, *CCR10*). When the patient received valganciclovir, flow cytometry analysis of PBMCs after 2 weeks revealed a reduction of CCR4${}^{+}$CCR10${}^{+}$CD4${}^{+}$, and CD8${}^{+}$ T cells.

### 3.5. Fibrosis

Fibrosis studies are generally focused on understanding the behavior of fibroblasts or immune populations. Kalekar et al. [63] studied the cell heterogeneity in lung and skin fibrosis. They observed that skin fibrosis is led by regulatory T cells similar to T helper cell type 2 (Th2) cells. This population expressing *GATA3* and *IRF4*—associated with Th2 cells—is different from the regulatory T cell population in the lung. Transcription factor analysis also supported that Th2 cells are enriched in skin immune populations in comparison to the lung. Therefore, it is likely that a cell population with a Th2-like phenotype is driving the skin fibrosis process.

### 3.6. Human Papillomavirus Iinfection (HPV)

Long-term immunosuppressive treatment leads to dramatically increased incidences of warts, anogenital neoplasias, and squamous skin cancers. Devitt et al. [64] biopsied three warts from an immunosuppressed organ recipient and detected an upregulation of markers of the altered skin barrier function (*ARL4A*, *MT2A*) and inflammation (*AP1*, *FOS*, and *JUN*). The third lesion showed increased expression of *KRT6A* and *MT1G* and may represent precancerous cells.

### 3.7. Keloid

Keloid has been studied using single-cell by Liu et al. [65]. In this study, single-cell pathway enrichment analysis revealed that *PDGF*, *NOTCH1*, and *Eph-Ephrin* pathways were enriched in keloid samples compared to controls. A secondary analysis, where cell–cell interactions are studied by measuring the coexpression of ligand-receptor pairs, found that *EFNB2*-*EPHA4*—between leukocytes, Schwann cells, and vascular cells—and *VEGFB*-*FLT1*—between leukocytes and vascular endothelial cells—pairs were enriched. Those results suggest a complex interaction network between nerve, vascular, and immune cell populations in the keloid. However, that picture is not complete because both keloid and scleroderma fibroblasts seem to specifically upregulate secretory proteins such as *POSTN*, *COMP*, and *ASPN* [66]. Of interest, Schwann cells also seem to contribute to keloid formation and thus may become a previously unrecognized player in this pathology [67]. The significant expansion of endothelial cells and, especially, collagen-expressing fibroblasts, observed by Liu et al. [65], had previously been observed in the central hypoxic part of keloids by Okuno et al. [68]. They reported a *VIM*${}^{+}$ population with an increased expression of autophagy and glycolysis genes, such as *LC3*, *HIF1A*, or *MCT4*.

### 3.8. Leprosy

Leprosy is an immune-mediated illness produced by Mycobacterium leprae. Two common leprosy types are tuberculoid leprosy (T-lep) in which the host controls the bacteria via antimicrobial activity mediated by T-cell release of GNLY or IL26 and lepromatous leprosy (L-lep) with numerous skin lesions and abundant bacilli. L-lep patients may undergo a reversal reaction (RR) in which patients change from L-lep to T-lep.

Ma et al. [69] analyzed samples from RR and L-lep patients, to discern key factors between RR and L- lep patients. RR samples showed an enrichment of *CD1A*${}^{+}$ dendritic cells, M1-like macrophages, and *GCMB*${}^{+}$*PRF1*${}^{+}$*GNLY*${}^{+}$ T cells, previously described as amCTL cells. On the other hand, L-lep samples contained a higher proportion of plasma cells, *IFN*${}^{+}$ macrophages (also observed by Hughes et al. [70]) and *TREM2*${}^{+}$ macrophages. Trajectory analysis of macrophages showed a transition between L-lep *TREM2*${}^{+}$ macrophages to RR M1 macrophages, passing through intermediary states, showing a possible transition mechanism in RR patients.

Keratinocytes from RR patients showed a proinflammatory signature (*ILG36*, *KLK5*, *KLK7*, *CX3CL1*, and *S100A2*), and interestingly, although L-lep samples showed a hyperproliferation of the spinous layer, they did not show any increase in a pro-inflammatory state. Lastly, fibroblasts from RR patients were enriched in *SFRP2*${}^{+}$ and *CXCL2*${}^{+}$ subtypes, which are also increased in inflamed states under other pathologies [40,70] and wound healing [71].

Finally, spatial transcriptomics of L-lep and RR lesions recapitulated the overall structure of granulomas in lepromatous patients. Both types of granulomas included a core of macrophages, surrounded by a mantle of lymphocytes. In RR granulomas, macrophages showed an antimicrobial profile by expression of *CCL3* and *CCL18*, as well as T cells, which expressed *GZMB*, *PRF1*, *CCL5*, and *CXCL1*. Surrounding the granuloma, dendritic cells and *CXCL12*${}^{+}$ pro-inflammatory fibroblasts were found, and near the epidermal layer, *SFRP2*${}^{+}$ fibroblasts —corresponding to clusters B2 and A1 of healthy fibroblasts in Ascension et al. [35]. Interestingly, both types of fibroblasts showed a transcriptomic core of antimicrobial response as well.

### 3.9. Melanoma

The hallmark study on melanoma was published by Tirosh et al. [72]. They analyzed the transcriptome of malignant and non-malignant cells in 19 tumors and observed that malignant cells would be either in the *MITF*${}^{+}$ (*MITF*${}^{+}$, *TYR*${}^{+}$, *PMEL*${}^{+}$) or the *AXL* (*AXL*${}^{+}$, *NGFR*${}^{+}$) programs. The main discovery in this paper was that expression patterns were not binary, i.e., tumors in the *MITF* or *AXL* program had a small population of cells with higher expression of *AXL* and *MITF*, respectively, not observed with bulk analysis. They observed that cells treated with RAF and MEK inhibitors—against the *MITF* program—showed a higher proportion of cells of the *AXL*${}^{+}$ program. Those tumors already had a small *AXL*${}^{+}$ population (<3%) before treatment, which favors a positive selection of the *AXL*${}^{+}$ population as resistant to the therapy. In addition, cancer-associated fibroblasts (CAFs) expressed *AXL*, showing that CAF abundance might be linked to preferential expression of the *AXL* over the *MITF* program. More recently, Tang et al. [73] studied melanocytes in areas of skin with different sun exposure patterns. Chronically exposed areas (face) had a lower mutation burden, i.e., higher adaptation–than intermittently sun-exposed areas (back). Many of the mutations found in healthy skin are weakly oncogenic.

### 3.10. Psoriasis

Psoriasis is a complex pathology that implies dysfunctional signaling between immune cell types and keratinocytes in the skin. Keratinocytes show a complex transcriptomic profile where *IL17*, *IL17RA*, and *IL17RC* [40,70] as well as common inflammatory markers *S100A7*, *S100A8*, and *S100A9* [40,43] are overexpressed in all skin layers (Figure 2). There is also a localized overexpression of *IFI27* and *PI3* in the suprabasal layer. Some of these results complement previous findings obtained by bulk microarray, GWAS, and NGS methods [56,59,74,75]. Such examples include the AD-like loss of differentiation signature from the top epidermal layers (*FLG*, *LOR*, *KRT10*, *KRT1*) and a loss of expression of genes involved in lipid metabolism (*EVOLV3*, *FABP5*) [74].

As it would be expected, Reynolds et al. found a decrease of basal markers and an increase of differentiation markers, implying a commitment of epidermal populations [40]. Moreover, keratinocytes also keep their hyperproliferative activity by overactivation of the urea cycle. *ASS1* activity—an arginine pool maintainer—was expressed, while no arginine transporter expression was observed, indicating a de novo arginine synthesis for urea cycle activity [76].

Apoptosis may also constitute a relevant factor in psoriasis, since autoreactive peripheral cells from psoriatic plaques show decreased cell apoptosis rates [77]. IFI27 is a protein that leads to sensitization to IFN-mediated extrinsic apoptosis [78,79]. Interestingly, it has been observed that lesional differentiated keratinocytes [40], as well as granular keratinocytes and melanocytes [43], showed an increased expression of *IFI27*. Gao et al. observed an increased regulation of apoptosis in epidermal basal cells [80]. These bulk transcriptomic studies await replication by single-cell analysis.

Immune populations are also important in the inflammatory progression of psoriasis, with a consensus that dendritic cell (DC) populations are important in this process. Cheng et al. [43] discovered a *CD1C*${}^{+}$*CD301A*${}^{+}$ DC population not previously described, and Hughes et al. [70] discovered an over-expression of *CCL17*, *CCL22*, and *IL12B* in an *IRF4*${}^{+}$ DC population. Gao et al. and Kim et al. use ligand-receptor analysis to show that DCs bind to T-cells, melanocytes, and suprabasal keratinocytes using *LILRB1* and *LILRB2* [80] and to pericytes, fibroblasts, and basal epidermis using *IL36G*, *WNT5A*, and *CD58* [81]. Most of these cell types overexpressed either MHC-I or MHC-II [81].

T cell populations are also key in the development of psoriasis, the most relevant being the CD8${}^{+}$ and CD17${}^{+}$. Penkava et al. [82] found a particular CD8${}^{+}$ T cell clonotype enriched in synovial fluid of patients with psoriatic arthritis. This clonotype, only found in synovial fluid and not in blood, expresses *CXCR3*, which binds to *CXCL9*, also only expressed in synovial fluid.

Liu et al. focused on CD8${}^{+}$ T cell heterogeneity during psoriasis [65]. They observed some clusters present in psoriatic samples, which showed a general overexpression of *CCR4*, *CCR8*, *CD69*, and *CXCL13*, previously reported by Tirosh et al. [72] to be associated with exhausted tumor-infiltrating lymphocytes. Among the clusters, Tc1-like (*IFGN*${}^{+}$, *TNF*${}^{+}$), cytotoxic (*GZMA*${}^{+}$, *GZMK*${}^{+}$), and early-activated (*CXCR3*${}^{\mathrm{hi}}$*CCR7*${}^{\mathrm{lo}}$) T cells were overrepresented.

Kim et al. determined a classification of T cells based on differential expression patterns of *IL17A* and *IL17F* [81]. On the one hand, *IL17A*${}^{+}$ cells expressed high levels of inflammatory cytokines, such as *IL26*, *CCL3*, *CCL4*, and *CCL5*; and *IL17A*${}^{+}$*IFNG*${}^{+}$ cells expressed TFs such as *RORC* or *STAT4*, inflammatory cytokines *IL36G* and *TNF*, and cytotoxic transcripts such as *GZMA*, *GZMB*. On the other hand, *IL17F*${}^{+}$*IL10*${}^{+}$ cells expressed *MAF*, *AHR*, *CD73*, and *IL1RN*; and *IL17F*${}^{+}$*IL10*${}^{-}$ cells expressed high levels of inflammatory cytokines *IL1B*, *IL2*, *IL24*, *IL34*, *EBI3* and *LTA*.

Reynolds et al. discovered an enrichment of T cells and type 2 macrophages, which expressed stress genes (*DNAJB1*, *HSPA1B*, *HSPA1A*, *JUN*), chemotactic molecules (CCL4L2, CCL4, CCL3L1, CCL3) and angiopoietin. This macrophage population decreased after 12-week methotrexate treatment [70].

### 3.11. Systemic Lupus Erythematosus (SLE)

SLE is characterized by the production of auto-reactive antibodies against nuclear antigens such as ribonucleoproteins, dsDNA, and histones. Fibrosis has been associated with poor response to treatment. Skin is a potential target for SLE analysis because it is readily available, unlike other affected organs such as the kidney. Der et al. [83] studied skin and kidney biopsies from SLE patients using scRNA-seq and discovered an increased IFN-I response both in skin keratinocytes and in kidney tubular cells. Specifically, keratinocytes showed a hyperproliferative response and an increase of *COL1A1* and *COL17A1* expression.

### 3.12. Wound Healing

Wound healing is a highly structured process that involves an active immune response and ECM remodeling. Many pathologies are related to dysfunctional wound healing, e.g., diabetes-associated ulcers, which makes this topic highly interesting to be approached by single-cell methods. However, most studies are currently performed in animal models of disease.

Lim et al. [84] analyzed why large wounds resolve with hair and without scar in mice, whereas smaller wounds resolve with a scar. They observed that *Gli1*, *Gli2*, *Ptch1*, and *Ptch2*—hedgehog components or activators—were expressed in the center of the large (regenerative) wounds by a subset of myofibroblasts. Those myofibroblasts also expressed dermal papilla markers *Hey1*, *Sema6a*, and *Wif1*, indicating hair regeneration in that area. In contrast, in small wounds, fibroblasts showed *Wnt2* and *Wnt10* expression, but no hedgehog activity. Therefore, wound-induced de novo hair follicle generation requires both *Wnt* and *hedgehog* activity. Joost et al. [85] observed that *Lgr5*${}^{+}$ cells, located in the hair follicle bulge, acquired an *Lgr6*${}^{+}$-like profile throughout the wound healing process. This process was mediated by an increase of interactions of *Lgr5*${}^{+}$ cells with stromal cells from the surroundings. Once *Lgr5*${}^{+}$ cells acquired *Lgr6*${}^{+}$ IFE-like transcriptome, both *Lgr5*${}^{+}$ and *Lgr6*${}^{+}$ cells migrated to the wound front. Guerrero-Juarez et al. [86] detected two activated fibroblast populations, which differentiated into an *Acta*${}^{+}$*Talgn*${}^{+}$ myofibroblast state. Some of those myofibroblasts were *Lyz2*${}^{+}$ which, using lineage tracing, were shown to derive from the myeloid origin and differentiate into de novo adipocytes.

Haensel et al. [71] detected, in a comparison of wounded and unwounded skin, three different populations enriched in wounds: a *Col17a*${}^{\mathrm{hi}}$ population enriched in stem cells and with quiescent phenotype, an early response (ER) population enriched in immediate response genes, and a growth arrest (GA) population expressing cycle arrest genes with hypoxia and Tnfα expression. In their analysis, they concluded that the *Col17a*${}^{\mathrm{hi}}$ population might activate into the ER population, which is necessary to transition into the GA population and react to stimuli in the migratory front.

Therefore, wound repair and scarring are orchestrated by many cell types, not only fibroblasts, but their specific effects are poorly understood [87]. More importantly, the clinical relevance of mouse work is uncertain, and human wound healing must be better understood. Pioneering clinical work by Theocharidis et al. found a *COL7A1*-expressing fibroblast population that was enriched in diabetic skin [88]. A combination of analyses led them to propose activation of *IL13* and *IFNG* as a potential therapy for diabetic foot ulcer (DFU) healing. In a parallel effort, the transcriptomic profiles of 384 individual cells obtained from two patients that presented diabetic and non-diabetic foot ulcers have been reported [89]. However, this work illustrated that extracting viable cells from the highly degraded tissue setting of a chronic ulcer may sometimes be challenging. Two recent preprints on single-cell-based characterization of DFUs and pressure ulcers seem to have overcome these limitations [90,91]. Theocharidis et al. now analyze 94,325 cells from 26-foot samples from both healing and non-healing DFUs [90]. They report that fibroblasts with the transcriptomic signature *MMP1*, *MMP3*, *IL6*, *CHI3L1*, *ASPN*, *POSTN*, *PLA2G2A*, and *TNFAIP6* are associated with good DFU healers. Of note, healing-associated fibroblasts present markers of healthy skin fibroblast subtypes B1 (*PLA2G2A*, *TNFAIP6*) and C3 (*ASPN*, *POSTN*) [35]. Li et al. analyze 1170 epidermal cells from the gluteal area of five pressure ulcer patients and report a novel *MHCII*${}^{+}$ keratinocyte population overrepresented in patients with impaired healing [91]. In all, these articles demonstrate the potential of single-cell-based approaches for the development of novel patient-tailored therapies, and specifically to induce successful wound healing in chronic ulcers.

## 4. Challenges for the Translation of Single-Cell Based Results to the Clinic

Despite the fact that single-cell analysis has provided an impressive range of discoveries of highly complex dermatological processes, translation of the obtained insight into novel diagnostics and therapeutics development is still in its infancy [90,91]. In this section, we propose some technical, biological, and human aspects that might underlie this “valley of death” phenomenon.

### 4.1. Technical Challenges

Single-cell technologies are considered fairly recent, and novelty in any research technique comes with the inherent price of unreliability. Currently, there is a vast number of single-cell methods in use, most of them developed at academic groups, and with relatively small interoperability between them. Lack of industrial standardization can pose a threat to the reproducibility of results. Commercial products are quickly becoming the new standards, although the price of reagents is still a relevant limiting step for the cost-effectiveness of these methods. Newer methods like *SPLIT-seq* [92] can increase the throughput—from 10,000 cells to 100,000 cells—but their protocols are harder to implement and still lack standardization.

Additionally, single-cell methods suffer from low coverage—few mRNA molecules are captured per cell—and high variability—gene expression between similar cell types can vary considerably. Both effects can be explained by transcriptional bursting [93] and technical artifacts of sample processing [94]. Some of these artifacts can be managed computationally, by transforming zero or low expression values based on the transcriptomic profile of similar cells [95,96]. However, those methods can also produce mathematical artifacts and hinder the reproducibility of detection of marker genes and downstream analysis of the samples [97].

Due to the high costs of single-cell methods, the experimental design should consider whether to favor (i) more coverage with fewer cells—resolving more specific and subtle transcriptional programs [98]—or (ii) a shallower coverage including more cells—providing more diverse subpopulations or capturing the rare ones [99]. Over the last few years, single-cell studies have tended to favor a higher number of cells [6,11], mainly due to the interest in finding rarer subpopulations (i.e., stem cells or cells in differentiating states) that tend to not appear with lower cell numbers. In clinical studies, however, it would be feasible to apply a previous filtering step—e.g., using Fluorescence-activated Cell Sorting (FACS) cell sorting and sequence fewer cells at higher coverage to detect less common transcriptional programs.

Lastly, another factor to consider in experimental design is the batch effect. Sample handling should be minimized and performed in similar conditions to reduce this effect. Otherwise, datasets with strong batch effects can produce artifacts of cell types from different batches appearing more transcriptomically similar than similar cell types between them. A similar effect is observed with dataset integration, i.e., combining datasets from several experiments to produce a unified dataset with an increased cell number [100]. Batch effect correction and dataset integration is actively being researched [99], and several methods have acceptable results [101]. Nonetheless, these tools perform best with datasets with distinct cell types shared across datasets, and tend to perform worse if datasets are more continuous, highly-detailed [102], or have different dataset qualities, or varied size proportions [103,104]. Of note, special care should be taken because batch correction may merge different cell types, or overcorrect *bona fide* biological variation (due to age, disease, etc.) [100,105,106].

### 4.2. Biological Challenges

A poor understanding of biological variability sources on single-cell datasets can bias observations towards false-positive results. Sample handling is a crucial aspect of any genomics method, but especially in single-cell, where data are so sensitive to individual cell changes. Time and conservation media are important modifiers of transcriptomic profiles, e.g., late sample handling can bias gene expression towards cold-shock regulators and reduce the cellular expression [107], which can be falsely interpreted as a dedifferentiation process. Ideally, samples should be processed before 2 h post-extraction, which requires specific biopsy processing protocols not easily implemented in clinical routines. Preservation medium is also a critical choice on single-cell analysis [108]. Fine-tuning of different media for each tissue is optimal to achieve proper sample processing. For example, Mirizio et al. [109] used *CryoStor CS10* in a pilot study with scleroderma samples and, although transcriptomic similarity with fresh samples was high in immune cells, fibroblasts and endothelial cells, the quality of keratinocyte and adipocyte populations was found to be compromised. Additionally, the skin presents a high ECM content and thus it is difficult to dissociate rapidly. Finally, the choice of isolating cells from epidermis, dermis, or whole skin preparations also requires fine-tuning of processing protocols.

An aspect that must be kept in mind is the potential lack of correlation of transcriptomic data with protein expression. Numerous post-translational modifications affect cellular physiology, and transcriptomic methods will be mostly blind to these. Therefore, if at all possible, analysis of the effector proteins should complement any mechanistic understanding emerging from single-cell RNA sequencing studies. The last biological aspect to be considered is sample heterogeneity, which is especially relevant for the different sources of human skin, as discussed previously [35,43].

### 4.3. Human Challenges

Similar to other experiments, human factors must be considered when designing any single-cell analysis. The hardest problems for the translation of single-cell methods are the difficulty of biopsy taking and the complexity of data analysis robustness. Biopsy taking of any affected area can be a traumatizing procedure for the patient, or even impossible depending on the location. Skin conditions offer some advantage on the location but biopsies are nonetheless disregarded unless completely necessary.

Additionally, single-cell data analysis is complex owing to the sparse nature of the datasets, i.e., gene expression is highly variable or even nonexistent. Correct analysis of these datasets requires biological expertise and an expert data analyst to understand dataset biases and take the correct decision on how to proceed. Incorrect analyses of datasets–especially large datasets requiring computationally intensive analyses–can result in incorrectly annotated populations or oversimplifications of complex cell states [33]. Moreover, translation of data analysis requires robust and standardized pipelines. Although the existence of software packages like *scanpy* or *Seurat*, broadly used by the single-cell community, helps to create standard pipelines [110,111], thresholding and QC/filtering decisions are usually arbitrarily decided, and far from being robustly implemented. Currently, some efforts into creating a start-to-end robust pipeline are being made [112], although these pipelines are not widely adopted by the single-cell community.

## 5. Opportunities for the Translation of Single-Cell Based Results to the Clinic

In this section, we propose possible novel diagnostic or therapeutic targets that can be readily translated to the clinic.

Regarding AD, there are several interesting targets. The *CCR7*/*CCL19* axis, which was upregulated in diseased samples [39], has previously been studied [113], and peptides for dimerization inhibition have already been produced [114]. *S100* and *CRIP1*, overexpressed in fibroblast populations described by He et al., are also interesting targets for inhibition. S100 family has already been associated with macrophage-mediated inflammation [115]. Amphiregulin is upregulated and promotes keratinocyte hyperproliferation, making inhibitors of EGF receptors putative targets for amphiregulin inhibition.

Two of the key pathways in the maintenance of keloids might be *NOTCH1* and *Eph-Ephrin* pathways. *NOTCH1* inhibitors [116,117], and *Eph-Ephrin* inhibitors [118,119], commonly used for cancer, might be of use in the initial stages of keloid.

*MITF*-*AXL* axis is key during melanoma development, and can easily condition the fate of targeted therapies. RAF and MEK inhibitors have shown a selection of the AXL${}^{+}$, MITF-resistant populations. AXL has been associated with a range of cancers with poor clinical outcomes [120]. Therefore, MITF- resistant populations could be targeted with AXL inhibitors, such as bemcentinib, currently in Phase II trials for different solid and hematological tumors. Moreover, the detection of circulating tumor cells with MITF/AXL bias might help optimize prognosis and treatment.

Psoriasis has been extensively studied using single-cell analysis, and there are several diagnostic and therapeutic targets for this disease. The detection of a *CD1C*${}^{+}$*CD301A*${}^{+}$ dendritic cell population in psoriatic samples by FACS might be feasible as a differential diagnostic method. Since de novo arginine synthesis is necessary by keratinocytes to keep the urea cycle active during hyperproliferation, ASS1 is a potential topical inhibition target to reduce hyperproliferation in psoriatic patients. Interestingly, ASS1 is a metabolic regulator of colorectal cancer pathogenesis [121], although the inhibitor used in the study was an shRNA, which opens new ways for ASS1 molecule inhibitors.

## 6. Concluding Remarks

We have shown that single-cell methods are very useful to unravel tissue heterogeneity or to find details on how a physiological or disease process develops. Single-cell discoveries have a high translation potential in areas like cancer, where circulating tumor cell signatures can help diagnose, stratify, and treat cancers that otherwise would be much harder to study. However, considering all the previously mentioned challenges (such as robustness of analyses), single-cell methods in their current state are hardly implementable into clinical applications, due to the technical limitations on the achievable information from single-cell technologies. Additionally, greater involvement of dermatologists alongside the biological community is needed to achieve the full clinical potential of these tools. Despite these limitations, single-cell technologies represent rather a useful tool for basic discoveries on homeostasis and disease that can be later used for diagnostic and treatment purposes (Figure 3).

Big efforts are being put into the generation of single-cell datasets with consistent protocols, looking for transparency and reproducibility, such as the Human Cell Atlas [122], Tabula Muris [123], or the Human Tumor Network Atlas [124]. The single-cell era is showing the potential of open science, and this, applied to skin knowledge, will be of paramount importance to develop new diagnostic and treatment tools for the clinic, based on the basic knowledge acquired from the single-cell community. Currently, there is an opportunity for researchers on skin conditions to broaden these efforts and make an impact on dermatological disease understanding.

## Figures and Tables

**Figure 1 life-12-00067-f001:**
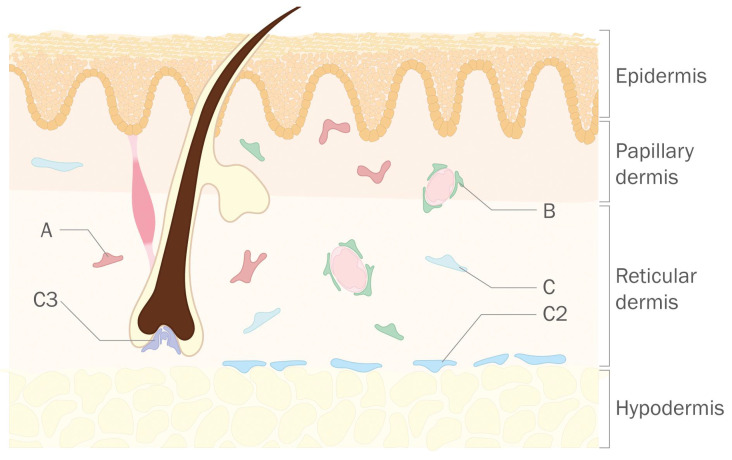
Localization of dermal fibroblast subpopulations as described by Ascensión et al. [33]. Type A, B and C cells are not compartmentalized to the papillary or reticular dermis. As observed from cell type markers and immunofluorescence assays, C3 fibroblasts might be associated with the hair follicle dermal papilla, C2 might interact with adipose cells from the dermal white adipose tissue (DWAT)/hypodermis, and B-type fibroblasts might interact with blood vessels.

**Figure 2 life-12-00067-f002:**
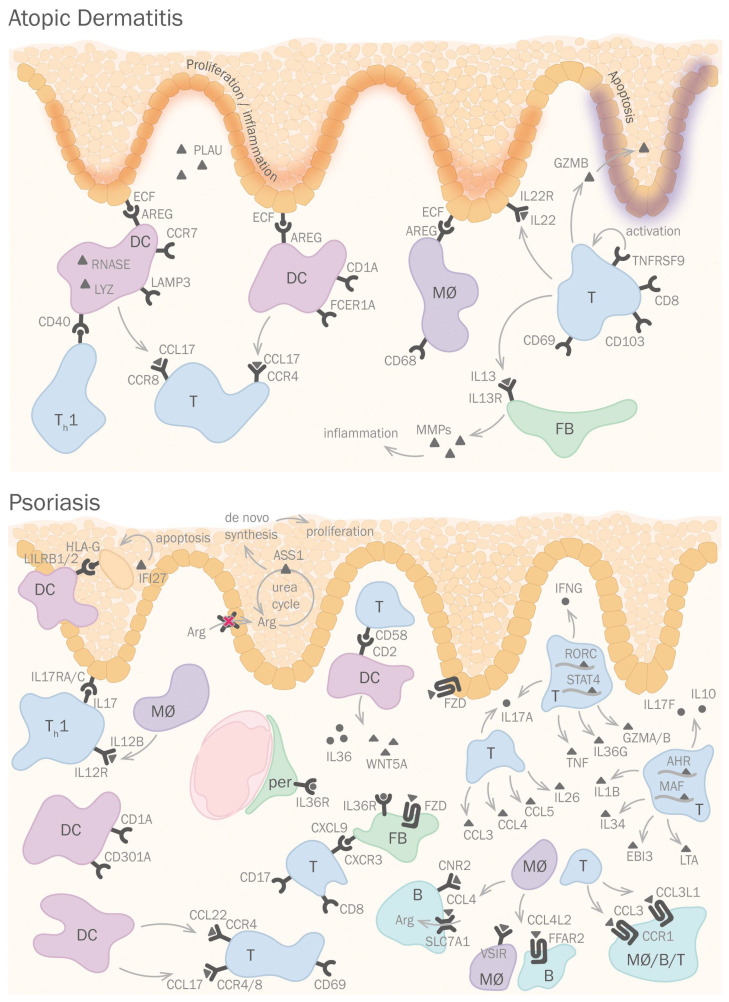
Single-cell analyses shed light on the intercellular signaling underlying dermatological disease. In AD (**top** panel), dendritic cells (DCs) and macrophages (MØ) interact with basal keratinocytes via amphiregulin (AREG) and PLAU secretion, enhancing keratinocyte proliferation. DCs also interact with T cells via CD40 and CCL17. T cells interact with fibroblasts via IL13, which enhances MMP secretion. In psoriasis (**lower** panel), Th and DCs interact with keratinocytes via IL17 and LILRB1/2. Additionally, IL36 and WNT5A secretion by DCs favors pericyte and fibroblast attraction. T cells secrete different chemotactic and inflammatory factors such as CCL3, CCL4, IL36G, IL1B, and EBI3. T cells and macrophages interact with other immune cells via CCL4, CCL4L2, CCL3, and CCL3L1. Nomenclature for the depicted cells is as follows: B—B cell, DC—dendritic cell, FB—fibroblast, MØ—macrophage, per—pericyte, T—T cell, Th—T helper cell.

**Figure 3 life-12-00067-f003:**
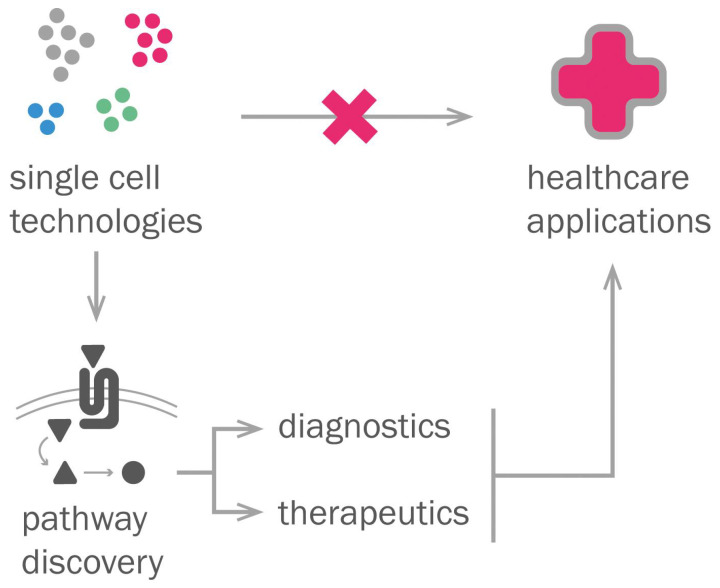
Healthcare applications of single-cell technologies. Single-cell technologies are unlikely to generate direct applications to the clinic. Instead, they might help the indirect development of healthcare applications by diagnostic and therapeutic target detection through lead pathway discovery.

## Data Availability

Not applicable.

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
