# Peer review of "Challenges and Opportunities for the Translation of Single-Cell RNA Sequencing Technologies to Dermatology"

_life, 2022, doi:10.3390/life12010067_

Round 1

Reviewer 1 Report

In this interesting review article, the authors summarized the use of single-cell RNA sequencing technologies in dermatology. I recommend the following:

1-The summary is too short, please add more phrases to your current abstract including the aim and implications.

2-In keloid, please add the role of autophagy genes in pathogenesis of this disease

3-In Psoriasis, please add the role of apoptosis-related genes in the pathogenesis of this disease

4-Add some paragraph related to  batch effect as  a common source of technical variation in high-throughput sequencing experiments 

5-Figure 1 is very nice, the types of fibroblasts and their role have significant implications in skin diseases. Can you add more roles for fibroblasts in  this figure using this reference: Okuno R, Ito Y, Eid N, Otsuki Y, Kondo Y, Ueda K. Upregulation of autophagy and glycolysis markers in keloid hypoxic-zone fibroblasts: Morphological characteristics and implications. Histol Histopathol. 2018 Oct;33(10):1075-1087. doi: 10.14670/HH-18-005. Epub 2018 May 29. PMID: 29809274

Author Response

In this interesting review article, the authors summarized the use of single-cell RNA sequencing technologies in dermatology. I recommend the following:

1-The summary is too short, please add more phrases to your current abstract including the aim and implications.

Thank you for your comments. We have extended the abstract to better reflect the aim and implications of the review.

2-In keloid, please add the role of autophagy genes in the pathogenesis of this disease

We added a statement about the upregulation of autophagy and glycolysis markers LC3, HIF1A, and MCT4 in hypoxic keloid. 

3-In Psoriasis, please add the role of apoptosis-related genes in the pathogenesis of this disease

We agree that a reanalysis of single-cell datasets focusing on this point would shed further light on the important role of apoptosis in psoriasis, but we could not find any such evidence in the literature. We have included a novel paragraph highlighting the existing evidence, which is based in bulk transcriptomics.  

4-Add some paragraph related to  batch effect as a common source of technical variation in high-throughput sequencing experiments 

We have added a novel paragraph regarding batch-effect correction in the Technical challenges section.

5-Figure 1 is very nice, the types of fibroblasts and their role have significant implications in skin diseases. Can you add more roles for fibroblasts in this figure using this reference: Okuno R, Ito Y, Eid N, Otsuki Y, Kondo Y, Ueda K. Upregulation of autophagy and glycolysis markers in keloid hypoxic-zone fibroblasts: Morphological characteristics and implications. Histol Histopathol. 2018 Oct;33(10):1075-1087. DOI: 10.14670/HH-18-005. Epub 2018 May 29. PMID: 29809274

We have added the relevant information in the review and quoted this reference. We respectfully submit that we don’t see this information fit for Figure 1, because it is focused on healthy skin and therefore information on keloid fibroblasts seems better suited for the Keloid section of the manuscript.

Reviewer 2 Report

The Authors give an extensive overview on the novel data that single cell RNAseq and Spatial transcriptomics is bringing to the field of various skin diseases. However, what I miss is exactly the dermatological aspect of the work. The review is likely to catch the attention of dermatologists, of whom many would like to see what is really new in understanding each disease. What did we not know before or perhaps what is the concept that got proven with these results? Are there any findings that could brake the ice or rather these data are interesting but have little of importance? Therefore, please include dermatologist(s) who could critically contribute to the manuscript and correct outdated statements such as “Thus, our current understanding of the skin at the cellular level is very distant from the classical textbook image of two layers mainly composed of keratinocytes and fibroblasts.” – I don’t think there are any textbook with such figure (the most widely used textbook, the Bolognia’s Dermatology, is nearly 3.000 pages, with dedicated parts to all cellular compartments of the skin).

Regarding the methodological introduction, Authors simply jumped over 15 years when whole tissue microarrays/RNAseq data delivered real cutting edge data. I’m especially interested in seeing what we missed in those whole tissue analyses and what we identified that was further confirmed with single cell RNAseq.

In the challenges part, the detection and thus the confirmation of the changes at the level of protein, which is very often missing from RNA studies, should be discussed as well.

Author Response

The Authors give an extensive overview of the novel data that single-cell RNAseq and Spatial transcriptomics is bringing to the field of various skin diseases. However, what I miss is exactly the dermatological aspect of the work. The review is likely to catch the attention of dermatologists, of whom many would like to see what is new in understanding each disease. What did we not know before or perhaps what is the concept that got proven with these results? 

Are there any findings that could break the ice or rather these data are interesting but have little importance? 

We thank the reviewer for this interesting point. Highlighting what is the new information that single-cell technologies have generated for each of the 12 diseases/conditions that we report would be an encyclopedic effort. The review is already 18 pages-long, which we find reasonable for the aim of highlighting how these techniques are opening up a new era of increased understanding in all of biology, with a particular focus on dermatology.  In any case, we made an effort to include previous bulk transcriptomic findings in some of the pathologies we addressed, such as psoriasis and atopic dermatitis.

Therefore, please include dermatologist(s) who could critically contribute to the manuscript and correct outdated statements such as “Thus, our current understanding of the skin at the cellular level is very distant from the classical textbook image of two layers mainly composed of keratinocytes and fibroblasts.” – I don’t think there is any textbook with such figure (the most widely used textbook, the Bolognia’s Dermatology, is nearly 3.000 pages, with dedicated parts to all cellular compartments of the skin).

With that statement, we meant to highlight that the heterogeneity of skin cell subpopulations is still to be reflected in Dermatology textbooks. However and for improved clarity, we have removed this phrase.

Regarding the methodological introduction, the Authors simply jumped over 15 years when whole tissue microarrays/RNAseq data delivered real cutting edge data. I’m especially interested in seeing what we missed in those whole tissue analyses and what we identified that was further confirmed with single-cell RNAseq.

As aforementioned, we fully agree with this comment but, in our opinion, this should wait for the writing of disease-focused reviews where specific pathways highlighted by previous genomic and transcriptomic studies and their correlation with newly baked single-cell data can be conveniently described and explored. Otherwise the length of this review would not be manageable. 

In the challenges part, the detection and thus the confirmation of the changes at the level of protein, which is very often missing from RNA studies, should be discussed as well.

This is also a good point. We have added a brief comment on the protein quantification aspects at the single-cell level in the introduction of the review. This comment reflects on the state-of-the-art of single-cell proteomic techniques, and how they are not suited yet to be used as a mainstream tool in single cell analysis. On the other hand, there is the aspect of the potential lack of correlation of transcriptomic data with protein expression. We have included this comment in the Biological Challenges section of the manuscript.

Reviewer 3 Report

 Overall, the content of this manuscript is comprehensive and the data reported may provide new challenges for the translation and application of single-cell based studies to the clinic, for a better management of patients.

I recommend the manuscript for publication. In the complex paper is well written, however I would suggest some minor revisions:

The potential usefulness of single-cell studies in the clinic is less described and explained.

The authors should also add some statements in in the paragraph “Opportunities for the translation of single-cell based results to the clinic” in order to summarize the current status of art of genomic studies, by NGS technologies, on personalized medicine in dermatological diseases such as psoriasis, atopic dermatitis and melanoma. (e.g. Pharmacogenetics and GWAS studies in psoriasis, genetic and epigenetic studies and therapy in Atopic dermatitis, Next Generation Sequencing Panel for Analysis of Circulating Tumor DNA in melanoma patient).

Author Response

Overall, the content of this manuscript is comprehensive and the data reported may provide new challenges for the translation and application of single-cell based studies to the clinic, for a better management of patients.

I recommend the manuscript for publication. In the complex paper is well written, however I would suggest some minor revisions:

The potential usefulness of single-cell studies in the clinic is less described and

explained.

Thank you. This comment is in line with previous reviewers. We have made an effort to better explain the usefulness of single-cell studies on top of previous bulk genomic and transcriptomic studies.  

The authors should also add some statements in in the paragraph “Opportunities for the translation of single-cell based results to the clinic” in order to summarize the current status of art of genomic studies, by NGS technologies, on personalized medicine in dermatological diseases such as psoriasis, atopic dermatitis and melanoma. (e.g. Pharmacogenetics and GWAS studies in psoriasis, genetic and epigenetic studies and therapy in Atopic dermatitis, Next Generation Sequencing Panel for Analysis of Circulating Tumor DNA in melanoma patient).

We agree; we have included novel paragraphs for these diseases in response to this comment and similar suggestions from the other reviewers.

Round 2

Reviewer 2 Report

Expert driven dermatological translation as suggested also by Reviewer 3 would have added significantly to the scientific merit of the work.

Author Response

Expert driven dermatological translation as suggested also by Reviewer 3 would have added significantly to the scientific merit of the work.

We agree with the reviewer that this would have added merit to the work, however, the relatively short times given for review of the manuscript made us take the choice of maintaining the authorship as original. We sincerely hope this relatively minor point does not preclude approval of the final version of the manuscript, which has been validated by researchers with many years of experience in cutaneous biology.